# ASYMMETRICALLY DECENTRALIZED FEDERATED LEARNING

## ABSTRACT

To address the communication burden and privacy concerns associated with the centralized server in Federated Learning (FL), Decentralized Federated Learning (DFL) has emerged, which discards the server with a peer-to-peer (P2P) communication framework. However, most existing DFL algorithms are based on symmetric topologies, such as ring and grid topologies, which can easily lead to deadlocks and are susceptible to the impact of network link quality in practice. To address these issues, This paper proposes the DFedSGPSM algorithm, which is based on asymmetric topologies and utilizes the Push-Sum protocol to effectively solve consensus optimization problems. To further improve algorithm performance and alleviate local heterogeneous overfitting in Federated Learning (FL), our algorithm combines the Sharpness Aware Minimization (SAM) optimizer and local momentum. The SAM optimizer employs gradient perturbations to generate locally flat models and searches for models with uniformly low loss values, mitigating local heterogeneous overfitting. The local momentum accelerates the optimization process of the SAM optimizer. Theoretical analysis proves that DFedSGPSM achieves a convergence rate of $\mathcal{O}(\frac{1}{\sqrt{T}})$ in a non-convex smooth setting under mild assumptions. This analysis also reveals that better topological connectivity achieves tighter upper bounds. Empirically, extensive experiments are conducted on the MNIST, CIFAR10, and CIFAR100 datasets, demonstrating the superior performance of our algorithm compared to state-of-the-art optimizers.

## 1 INTRODUCTION

Federated Learning (FL) is an emerging distributed machine learning paradigm that ensures privacy protection (McMahan et al., 2017). It allows multiple participants to collaboratively train models without sharing their raw data. Currently, most research (Acar et al., 2021; Zhou & Li, 2023; Sun et al., 2023) efforts have focused on Centralized Federated Learning (CFL). However, the existence of a central server in CFL introduces challenges such as communication burden, single point of failure (Chen et al., 2023) and privacy leaks (Gabrielli et al., 2023). In comparison, Decentralized Federated Learning (DFL) offers enhanced privacy protection (Cyffers & Bellet, 2022), faster model training (Lian et al., 2017), and robustness towards slow client devices (Neglia et al., 2019). Therefore, DFL has emerged as a popular alternative solution at present (Chen et al., 2023).

Currently, DFL can be categorized into two types based on communication topology: symmetric topology and asymmetric topology, represented by undirected and directed graphs, respectively. Extensive research has been conducted on symmetric DFL(Shi et al., 2023; Li et al., 2023; Lian et al., 2017). However, symmetric DFL still faces some challenges in practical applications. Fristly, (Tsianos et al., 2012) have studied the practical scenarios of symmetric communication, which is prone to deadlocks, leading to system stagnation. Secondly, in the actual DFL process, each client can ensure that the weighted coefficients of the information it sends are 1. However, due to network communication quality, it cannot be guaranteed that the sum of the weighted coefficients of the information received by each client is 1 (Zeng & Yin, 2017). This data loss caused by communication quality not only disrupts the assumption of symmetric DFL with symmetric doubly stochastic mixing matrices(Sun et al., 2022) but also makes the algorithm difficult to optimize or even converge. Finally, (Zhang & You, 2019) mentioned that many applications require directed communication networks, where upper-level clients can send data information to lower-level clients, but not vice

versa. We found that the culprit causing these issues is the assumption of symmetry. However, simply changing the restriction of the mixing matrices from symmetric to asymmetric introduces additional biases in the optimization process. Specifically, in the case of undirected graph mixing matrices, the matrix representation of directed graphs is not a doubly stochastic matrix. This means that in the mixing matrix $[p_{i,j}]_{n \times n}$ of the directed graph, $\sum_{j=1}^{n} p_{i,j} \neq 1$. If each client directly aggregates using the method $\mathbf{x}_i = \sum_{j=1}^{n} p_{i,j} \mathbf{x}_j$ (Shi et al., 2023), it will introduce additional bias.

To address the potential bias introduced by asymmetric communication, our algorithm DFedSG-PSM incorporates the ideas from the Push-Sum (Assran et al., 2019; Kempe et al., 2003) algorithm. Specifically, each client $i$ calculates the PUSH-SUM weight $w_i = \sum_{j=1}^{n} p_{i,j} w_j$, and obtains the unbiased parameter $\mathbf{x}_i / w_i$ for subsequent training processes. Furthermore, to alleviate the issue of local heterogeneous overfitting in FL and further improve algorithm performance, our algorithm combines the SAM optimizer and local momentum. SAM generates locally flat models through gradient perturbations, which offers two advantages. Firstly, gradient perturbations effectively prevent local overfitting. Secondly, studies (Zhong et al., 2022) have shown that generating locally flat models by each client can lead to a relatively flat landscape of the aggregated global model, thereby improving generalization capability. Additionally, utilizing local momentum helps the SAM optimizer to quickly find locally flat models, reducing the required communication rounds for convergence.

Theoretically, we have proven that the DFedSGPSM algorithm converges at a rate of $\mathcal{O}\left(\frac{1}{\sqrt{T}}\right)$ in a non-convex setting. The theoretical results indicate that as the connectivity of the communication topology improves, the convergence upper bound becomes tighter. On the other hand, we extend the results of (Chen et al., 2023) by setting $\rho = 0$ and $\alpha = 0$ in Algorithm 1. While (Chen et al., 2023) only proved the convergence of the algorithm in a strongly convex setting, we weaken the assumption of strong convexity to non-convexity. Empirically, we conducted extensive tests on MNIST, CIFAR10/100 datasets. The experimental results demonstrate that our algorithm significantly outperforms the current state-of-the-art symmetric DFL methods such as DFedSAM (Shi et al., 2023) in terms of both generalization performance and convergence speed.

In summary, our main contributions are four-fold:

- We propose DFedSGPSM, a novel asymmetric DFL algorithm. By integrating the idea of the Push-Sum algorithm, we effectively reduce the introduction of additional bias during the aggregation process. Furthermore, our algorithm combines the SAM optimizer and local momentum to alleviate local heterogeneous overfitting in FL, achieving to improved generalization performance.

- We propose DFedSGPSM-S, which allows clients to choose neighboring clients for sending model parameters, using a neighbor selection strategy. This strategy enhances flexibility in our asymmetric DFL framework and is better suited for real-world applications compared to symmetric DFL.

- We derive theoretical analysis for DFedSGPSM, which demonstrates $\mathcal{O}\left(\frac{1}{\sqrt{T}}\right)$ convergence rate in non-convex smooth settings. Furthermore, the convergence bound of the algorithm becomes tighter as the connectivity of the communication topology improves.

- We conducted extensive experiments on MNIST, CIFAR10/100 datasets to validate the performance of our algorithm. The results demonstrate that our approach achieves SOTA performance in terms of both convergence speed and generalization ability under general federated settings.

## 2   RELATED WORK

In this paper, we briefly review the three main lines of work that are most relevant to our research: Sharpness-Aware Minimization (SAM), Momentum, and Decentralized Federated Learning (DFL).

**Decentralized Federated Learning.** To address the communication burden on the server in centralized scenarios, decentralized communication methods can distribute the communication load to each node while keeping the overall communication complexity the same as in centralized scenarios (Lian et al., 2017). Additionally, decentralized communication methods can better protect privacy (Yang et al., 2019; Lalitha et al., 2018; 2019). In symmetric DFL, (Sun et al., 2022) extended the FedAvg (McMahan et al., 2017) algorithm to decentralized scenarios and combined it with momentum acceleration to improve convergence. (Dai et al., 2022a) introduced sparse training into DFL to

reduce communication and computation costs. (Shi et al., 2023) applied SAM to DFL and enhanced the consistency among clients by incorporating Multiple Gossip Steps. In asymmetric DFL, (Chen et al., 2023) applied the Push-Sum algorithm to DFL and proposed a neighbor selection strategy that accelerates training speed and reduces communication consumption. For more related work on DFL, please refer to the survey papers (Gabrielli et al., 2023; Yuan et al., 2023; Beltrán et al., 2022).

**Momentum.** Momentum is a commonly used acceleration technique in deep learning (Sutskever et al., 2013). In CFL, FedCM (Xu et al., 2021) estimates the global momentum at the server and applies it to each client's update, which mitigates the issue of client heterogeneity. Similarly, (Karimireddy et al., 2020) propose MimeLite, which also utilizes global momentum in CFL. Additionally, (Wang et al., 2019) introduce SLOMO, which applies an additional momentum step to the average model parameters in each round of communication to improve the convergence of distributed training. It updates momentum based on the differences between consecutive server model parameters. In DFL, (Sun et al., 2022) utilize local momentum at the client-side to reduce the required communication rounds for convergence. QG-DSGDm (Lin et al., 2021) is a momentum-based decentralized optimization method, where each client calculates a global momentum by using the averaged models sent by their neighboring clients. This effectively reduces client heterogeneity issues.

**Sharpness-Aware Minimization(SAM).** SAM (Foret et al., 2020) is a powerful optimizer in deep learning that enhances the model's generalization capabilities by finding the flat geometry of the loss landscape. SAM and its variants have been successfully applied to various machine learning tasks (Zhong et al., 2022; Zhao et al., 2022b; Kwon et al., 2021; Du et al., 2021). Recently, (Shi et al., 2023) successfully applied SAM to the field of symmetric DFL and proposed DFedSAM, achieving state-of-the-art (SOTA) performance. Additionally, in the CFL field, (Sun et al., 2023) combined SAM with the proximal term and its modification, also achieving SOTA performance.

The work most relevant to ours is DFedSAM by (Shi et al., 2023), which introduced the SAM optimizer into symmetric DFL to alleviate local heterogeneous overfitting and achieved state-of-the-art results. In our algorithm, we extend DFedSAM to a more general and practical asymmetric DFL setting. Additionally, we incorporate local momentum to accelerate the SAM optimizer's search for locally flat models. Another related work is AsyNG by (Chen et al., 2023), which combines Push-Sum and Neighbor selection strategies to ensure convergence in asymmetric DFL under the strong convexity assumption. In our algorithm, we expand on their work to handle more general non-convex settings. Additionally, we consider the issue of local heterogeneous overfitting in FL and mitigate it by combining the SAM optimizer with local momentum.

## 3 METHODOLOGY

In this section, we will present the necessary background information and introduce our proposed method. We will provide explanations for the implicit meanings of each variable and provide a detailed demonstration of the inference process of the DFedSGPSM algorithm.

### 3.1 NOTATIONS AND PRELIMINARY.

Let $n$ be the total number of clients. $T$ represents the number of communication rounds. $(\cdot)_{i,k}^t$ indicates variable $(\cdot)$ at the $k$-th iteration of the $t$-th round in the $i$-th client. $\mathbf{x}$ denotes the model parameters. $\mathbf{g}$ represents the stochastic gradient computed using the sampled data in Algorithm1. $p_{i,j}$ represents the weight of the link from client $j$ to client $i$. $w$ represents the PUSHSUM weight. The inner product of two vectors is denoted by $\langle \cdot, \cdot \rangle$, and $\|\cdot\|$ represents the Euclidean norm of a vector. Other symbols have their definitions provided in the respective references.

### 3.2 PROBLEM SETUP

We consider a network of $n$ nodes whose goal is to solve distributedly the following minimization problem:

$$\min_{\mathbf{x} \in \mathbb{R}^d} f(\mathbf{x}) := \frac{1}{n} \sum_{i=1}^{n} f_i(\mathbf{x}), \ \ f_i(\mathbf{x}) = \mathbb{E}_{\xi \sim \mathcal{D}_i} F_i(\mathbf{x}; \xi), \tag{1}$$

where only client $i$ knows the non-convex function $f_i : \mathbb{R}^d \to \mathbb{R}$ and $\mathcal{D}_i$ represents the data distribution in the $i$-th client, which exhibits heterogeneity across clients. The total number of clients

is denoted by $n$. Each client's local objective function $F_i(\mathbf{x}; \xi)$ is associated with the training data samples $\xi$. Let $f^*$ represent the minimum value of $f$, where $f(x) \geq f(x^*) = f^*$ for all $x \in \mathbb{R}^d$.

## 3.3 DIRECTED COMMUNICATION NETWORK

In the decentralized directed communication network, which refers to an asymmetric topology, clients communicate with each other in a peer-to-peer (P2P) manner. In a more general sense, the time-varying communication network between clients can be represented as a directed connected graph denoted as $\mathcal{G}(t) = (\mathcal{N}, \mathcal{E}(t), \mathbf{P}(t))$, where $\mathcal{N} = \{1, 2, \dots, n\}$ represents the set of clients, $\mathcal{E}(t) \subseteq \mathcal{N} \times \mathcal{N}$ denotes the links between clients, and $(i, j) \in \mathcal{E}(t)$ represents a directed link from client $i$ to client $j$. Additionally, we introduce the notation $N_i^{in}(t) = \{j \mid (j, i) \in \mathcal{E}(t)\} \cup \{i\}$ and $N_i^{out}(t) = \{j \mid (i, j) \in \mathcal{E}(t)\} \cup \{i\}$ for the in-neighbors and out-neighbors of client $i$, respectively, at time $t$. Additionally, We require these neighborhoods to include the node $i$ itself[1].

Compared to undirected graphs, directed graphs have the following advantages in DFL: (1) Undirected graphs are more prone to deadlock in practical applications (Tsianos et al., 2012) . In contrast, directed graphs can mitigate this issue by flexibly selecting neighbors within clients. (2) Directed graphs are more general than undirected graphs, as an undirected graph can be seen as a special case of a directed graph when every node in the directed graph satisfies $N_i^{in}(t) = N_i^{out}(t)$, for all $i \in \mathcal{N}$. (3) Directed graphs exhibit higher robustness in terms of network communication quality. In undirected graphs, if a communication link in one direction is disrupted, the symmetry of the undirected graph is broken, leading to a performance decrease in the entire DFL system. However, this situation does not occur in directed graphs (Zhang & You, 2019).

## 3.4 DFEDSGPSM ALGORITHM

In this section, we will introduce our proposed method to mitigate the negative impact of heterogeneous data and reduce the number of communication rounds in a directed communication network.

Our proposed DFedSGPSM is shown in Algorithm 1. At the beginning of each round $t$, Each local client performs five stages for K iterations: (1) Compute the debiased parameter $\mathbf{z}_{i,k+1}^t$ by dividing the local client model parameter $\mathbf{x}_{i,k}^t$ by the PUSHSUM weight $w_i^t$; (2) computing the unbiased stochastic gradient $\mathbf{g}_{i,k,1}^t = \nabla f_i(\mathbf{z}_{i,k+1}^t; \varepsilon_{i,k}^t)$ with a randomly sampled mini-batch data $\varepsilon_{i,k}^t$ and executing a gradient ascent step in the neighbourhood to approach $\breve{\mathbf{z}}_{i,k+1}^t$; (3) computing the unbiased stochastic gradient $\mathbf{g}_{i,k+1}^t$ with the same sampled mini-batch data in (2) at the $\breve{\mathbf{z}}_{i,k+1}^t$ to introduce a basic perturbation to the vanilla descent direction; (4) using the perturbed gradient $\mathbf{g}_{i,k+1}^t$ to compute the momentum-term $\mathbf{v}_{i,k+1}^t$; (5) executing the gradient descent step with the momentum-term. After $K$ iterations local training, The current local offset $(\mathbf{x}_{i,K}^t - \mathbf{x}_{i,0}^t)$ will be updated as the exponentially weighted sum of historical perturbed gradients, as shown in Equation 3. Then each local client selects its out-neighbors and sends $\left(p_{j,i}^t \mathbf{x}_i^{t+\frac{1}{2}}, p_{j,i}^t w_i^t\right)$ to them and receives $\left(p_{i,j}^t \mathbf{x}_j^{t+\frac{1}{2}}, p_{i,j}^t w_j^t\right)$ from in-neighbors. Finally, the client updates its model parameters and PUSHSUM weight by aggregating the parameter information received from its in-neighbors.

**Sharpness-Aware Minimization(SAM) and Momentum.** The SAM optimizer primarily enhances the generalization ability of the model by searching for the flat geometry of the loss landscape through gradient perturbation (Foret et al., 2020), which makes the loss landscape of the model parameters aggregated from neighboring models relatively flat. Unlike the gradients used in the vanilla SGD method, The SAM optimizer calculates the perturbed gradients by additionally computing a gradient ascent. Many studies (Foret et al., 2020; Mi et al., 2022; Zhao et al., 2022a; Zhong et al., 2022) have already pointed out that such an extra gradient ascent in the neighborhood can effectively capture the curvature near the current parameters, thereby enhancing generalization performance. The three-step process of the SAM optimizer can be found in lines 6-8 of Algorithm 1. Additionally, our proposed algorithm incorporates a momentum term $\mathbf{v}_{i,k}^t$ after computing the perturbed gradient, which helps the SAM optimizer to more quickly find locally flat models. The specific update equation can be found in lines 10 and 11 of Algorithm 1.

---

[1]Alternatively, one may define these neighborhoods in a standard way of the graph theory, but require that each graph in the sequence $\{\mathcal{G}(t)\}$ has a self-loop at every node.

---

**Algorithm 1** DFedSGPSM Algorithm Framework

---

**Input:** Initialzie $\eta_l > 0$, $\mathbf{x}_i^0 = \mathbf{z}_i^0 \in \mathbb{R}^d$ and $w_i^0 = 1$, $p_{j,i}^t = \frac{1}{|N_i^{out}(0)|}$ for all nodes $i, j \in \{1, 2, \dots, n\}$

**Output:** model average parameters $\bar{\mathbf{x}}^t$.

1: **for** $t = 0, 1, 2, \cdots, T - 1$ **do**
2:      **for** client $i$ in parallel **do**
3:         set $\mathbf{x}_{i,0}^t = \mathbf{x}_i^t$ and $\mathbf{v}_{i,0}^t = 0$
4:         **for** $k = 0, 1, 2, \cdots, K - 1$ **do**
5:            $\mathbf{z}_{i,k+1}^t = \mathbf{x}_{i,k}^t / w_i^t$
6:            sample a minibatch $\varepsilon_{i,k}^t$ and do
7:            compute unbiased stochastic gradient: $\mathbf{g}_{i,k,1}^t = \nabla f_i(\mathbf{z}_{i,k+1}^t; \varepsilon_{i,k}^t)$
8:            update extra step: $\breve{\mathbf{z}}_{i,k+1}^t = \mathbf{z}_{i,k+1}^t + \rho \frac{\mathbf{g}_{i,k,1}^t}{\|\mathbf{g}_{i,k,1}^t\|}$
9:            compute unbiased stochastic gradient: $\mathbf{g}_{i,k+1}^t = \nabla f_i(\breve{\mathbf{z}}_{i,k+1}^t; \varepsilon_{i,k}^t)$
10:           update the momentum step: $\mathbf{v}_{i,k+1}^t = \alpha \mathbf{v}_{i,k}^t + \mathbf{g}_{i,k+1}^t$
11:           update the gradient descent step: $\mathbf{x}_{i,k+1}^t = \mathbf{x}_{i,k}^t - \eta_l^t \mathbf{v}_{i,k+1}^t$
12:         **end for**
13:         $\mathbf{x}_i^{t+\frac{1}{2}} = \mathbf{x}_{i,K}^t$
14:         Send $\left(p_{j,i}^t \mathbf{x}_i^{t+\frac{1}{2}}, p_{j,i}^t w_i^t\right)$ to out-neighbors;
           receive $\left(p_{i,j}^t \mathbf{x}_j^{t+\frac{1}{2}}, p_{i,j}^t w_j^t\right)$ from in-neighbors
15:         $\mathbf{x}_i^{t+1} = \sum_j p_{i,j}^t \mathbf{x}_j^{t+\frac{1}{2}}$
16:         $w_i^{t+1} = \sum_j p_{i,j}^t w_j^t$
17:      **end for**
18: **end for**

---

It is worth noting that our de-biasing step is performed within the client update loop. This means that for each update of the client's parameters $\mathbf{x}_{i,k}^t$, the de-biased parameters $\mathbf{z}_{i,k}^t$ are also updated. In contrast, in the algorithm 1 proposed by (Chen et al., 2023), the de-biasing step is executed during the communication process with neighboring clients. Additionally, in Chen et al.'s algorithm 1, the gradients used during the $\tau$ iterations of client update loop are always different mini-batch stochastic gradients at the same de-biased parameter. This is another difference between the two algorithms.

## 4    CONVERGENCE ANALYSIS

In this section, we will propose some necessary assumptions for the convergence of the DFedS-GPSM algorithm and provide a detailed convergence theorem for our proposed algorithm. The detailed derivation process can be found in the appendix B.

**Assumption 1** ($\mathcal{B}$-**bounded strongly connected**) *The time-varying graph (i.e., the communication topology) is defined as $\mathcal{B}$-bounded strongly connected. This means that there exists a window size $\mathcal{B} \geq 1$ such that the union of graphs $\bigcup_{k=l}^{l+\mathcal{B}-1} \mathcal{G}(k)$ is strongly connected, where $l = 0, 1, 2, \cdots$.*

**Assumption 2** (**L-Smoothness**) *The non-convex function $f_i$ satisfies the smoothness property for all $i \in [m]$, i.e., $\|\nabla f_i(\mathbf{x}) - \nabla f_i(\mathbf{y})\| \leq L\|\mathbf{x} - \mathbf{y}\|$, for all $\mathbf{x}, \mathbf{y} \in \mathbb{R}^d$.*

**Assumption 3** (**Bounded Stochastic Gradient**) *The stochastic gradient $\mathbf{g}_{i,k}^t = \nabla f_i(\mathbf{x}_{i,k}^t, \varepsilon_{i,k}^t)$ with the randomly sampled data $\varepsilon_{i,k}^t$ on the local client $i$ is unbiased and with bounded variance, i.e., $\mathbb{E}[\mathbf{g}_{i,k}^t] = \nabla f_i(\mathbf{x}_{i,k}^t)$ and $\mathbb{E}\|\mathbf{g}_{i,k}^t - \nabla f_i(\mathbf{x}_{i,k}^t)\|^2 \leq \sigma_l^2$, for all $\mathbf{x}_{i,k}^t \in \mathbb{R}^d$.*

**Assumption 4** (**Bounded Heterogeneity**) *The dissimilarity of the dataset among the local clients is bounded by the local and global gradients, i.e., $\mathbb{E}\|\nabla f_i(\mathbf{x}) - \nabla f(\mathbf{x})\|^2 \leq \sigma_g^2$, for all $\mathbf{x} \in \mathbb{R}^d$. This paper also assumes global variance is bounded, i.e., $\frac{1}{m} \sum_{i=1}^m \|\nabla f_i(\mathbf{x}) - \nabla f(\mathbf{x})\|^2 \leq \sigma_g^2$.*

**Assumption 5** (**Bounded gradient**) *we have $\|\nabla f_i(\mathbf{x})\| \leq B$. for any $i \in \{1, 2, \cdots, m\}$.*

Assumption 1 is a typical assumption commonly adopted in prior work (Nedić & Olshevsky, 2014; Chen et al., 2023): it is considerably weaker than requiring each $\mathcal{G}(t)$ be connected for it allows the

edges necessary for connectivity to appear over a long time period and in arbitrary order. Assumption 2–5 are mild and commonly used in characterizing the convergence rate of DFL (Li et al., 2023).

There are three challenges at the theoretical analysis level: (1) Compared to the classical PUSHSUM method SGP(Assran et al., 2019), the main difficulty arises from the fact that after multiple local iterations, $\mathbf{x}_{i,K}^t - \mathbf{x}_{i,0}^t$ is not an unbiased estimate of $\nabla f(\mathbf{x}_i^t)$. Therefore, multiple local iterations are not trivial; (2) In contrast to the symmetric topology, our proposed algorithm employs an asymmetric topology, leading to the sum of $\sum_j p_{i,j}^t$ not being equal to 1. As a consequence, each client needs to maintain a PUSHSUM weight $w_i^t$ to de-bias the model parameters. This additional de-biasing step poses further challenges for theoretical analysis. (3) To obtain more general results, we conduct our analysis in the non-convex setting instead of the convex setting (Chen et al., 2023). Next, let $\bar{\mathbf{x}}^t = \frac{1}{n}\sum_{i=1}^n \mathbf{x}_i^t$. we will prove the convergence of the algorithm proposed in terms of $\bar{\mathbf{x}}^t$.

**Theorem 1** *Under the Assumptions 1–5, Let $f^* = inf_{\mathbf{x}} f(\mathbf{x})$ and assume $f^* > -\infty$. There exist constants $C > 0, q \in [0,1)$ and $\delta > 0$, when the local learning rate $\eta_l$ satisfies $0 < \eta_l < \frac{\delta}{8LK}$ and selects the proper values of $\rho$, then the sequence $\{\bar{\mathbf{x}}^t\}_{t\geq 0}$, generated by executing Algorithm 1, satisfies:*

$$\frac{1}{T}\sum_{t=0}^{T-1}\mathbb{E}\|\nabla f(\bar{\mathbf{x}}^t)\|^2 \leq \frac{2(1-\alpha)\left(f(\bar{\mathbf{x}}^0)-f^*\right)}{T\eta_l K} + \frac{L\eta_l K\sigma_l^2}{1-\alpha} + \frac{3\eta_l^2 L^2}{\delta}\left(C_1 + 64K^2B^2\right)$$

$$+ \frac{6L^2C^2}{T(1-q^2)}\frac{1}{n}\sum_{i=1}^n\|\mathbf{x}_i^0\|^2 + 6L^2\eta_l^2K^2C^2\frac{(B^2+\sigma_l^2)}{(1-\alpha)^2(1-q)^2} + 3L^2\rho^2$$

*Where $C_1 := 8K\sigma_l^2 + 64K^2\sigma_g^2 + \frac{32K^2\alpha^2}{(1-\alpha)^2}(\sigma_l^2 + B^2) + 64KL^2\rho^2$, the parameters $C$ and $q$ are related to the communication topology, and their specific definitions can be referred to as Lemma 3 in (Assran et al., 2019). $\delta$ represents the minimum sum of any row elements in the matrix $\prod_{i=1}^t \mathcal{G}(i)$ for all $t \geq 0$. Its specific definition can be found in Proposition 2.1 in (Taheri et al., 2020).*

For more detailed proof, please refer to the **Appendix**. With Theorem 1, we can state the following convergence rates for DFedSGPSM algorithms.

**Corollary 1** *Let the local adaptive learning rate satisfy $\eta_l = \mathcal{O}(\frac{1}{LK\sqrt{T}})$ and the perturbation parameter $\rho = \mathcal{O}(\frac{1}{L\sqrt{T}})$. Then, the convergence rate for DFedSGPSM satisfies:*

$$\frac{1}{T}\sum_{t=0}^{T-1}\mathbb{E}\|\nabla f(\bar{\mathbf{x}}^t)\|^2 = \mathcal{O}\Big(\frac{L(1-\alpha)\left(f(\bar{\mathbf{x}}^0)-f^*\right)}{\sqrt{T}} + \frac{\sigma_l^2}{\sqrt{T}(1-\alpha)} + \frac{C_2}{T\delta} + \frac{L^2C^2}{T(1-q^2)}\frac{1}{n}\sum_{i=1}^n\|\mathbf{x}_i^0\|^2$$

$$+ \frac{C^2(B^2+\sigma_l^2)}{T(1-\alpha)^2(1-q)^2} + \frac{1}{T}\Big).$$

*Where $C_2 = \frac{\sigma_l^2}{K} + \sigma_g^2 + \frac{\alpha^2}{(1-\alpha)^2}(\sigma_l^2 + B^2) + \frac{1}{TK}$. In particular, by setting $\rho = 0$ and $\alpha = 0$, we obtain the convergence of the AsyNG algorithm (Chen et al., 2023) in non-convex setting. This extends the results of (Chen et al., 2023), which assumed that the loss functions on each client are strongly convex.*

**Remark 1** *According to the results of (Assran et al., 2019), the value of $q$ in Corollary 1 can be explicitly expressed as $q = (1 - a^{\Delta \mathcal{B}})^{\frac{1}{\Delta \mathcal{B}+1}}$. where $\Delta$ represents the diameter of the communication topology, $\mathcal{B}$ has the same meaning as in Assumption 1, and $a < 1$ is a constant. It is evident that as the connectivity of the communication network improves, the value of $q$ will become smaller. According to Corollary 1, this will effectively improve the convergence bound of our algorithm. This conclusion is consistent with the results of (Li et al., 2023; Sun et al., 2022; Shi et al., 2023) in the symmetric DFL. Additionally, the value of $C$ decreases as the connectivity of the communication network improves, leading to the same conclusion. For specific details, please refer to Lemma 3 in the work of (Assran et al., 2019). We will not elaborate further on this point here.*

**Remark 2** *From Corollary 1, it can be seen that our proposed algorithm, DFedSGPSM, has a convergence rate of $\mathcal{O}\left(\frac{1}{\sqrt{T}}\right)$. This result is consistent with the convergence rate achieved by (Li et al., 2023; Sun et al., 2022; Shi et al., 2023) in the symmetric DFL. Furthermore, our results obtained*

*in the non-convex setting are more comprehensive compared to the results of (Chen et al., 2023), who achieved $\mathcal{O}(\frac{1}{T^\theta}), \theta \in (0,1)$ in the strongly convex setting. Additionally, when the smoothness is poor, i.e., when $L$ is large, the convergence bound will become looser. Finally, the term $\mathcal{O}\left(\frac{1}{T} + \frac{1}{T^2 K\delta}\right)$ can be neglected compared to $\mathcal{O}\left(\frac{1}{\sqrt{T}}\right)$, which arises from the additional SGD step for smoothness via the SAM local optimizer.*

## 5 EXPERIMENTS

In this section, we will evaluate the effectiveness of our proposed algorithm compared to seven baselines from CFL, symmetric DFL and asymmetric DFL. We present the convergence and generalization performance in Section 5.2, and study ablation experiments in Section 5.3.

### 5.1 EXPERIMENT SETUP

**Dataset and Data Partition.** The effectiveness of the proposed DFedSGPSM methods is evaluated on the MNIST, CIFAR-10, and CIFAR-100 datasets (Krizhevsky et al., 2009) in both IID and non-IID settings. To simulate non-IID data distribution among federated clients, the Dirichlet Partition approach (Hsu et al., 2019) is utilized. Specifically, the local data of each client is partitioned by sampling label ratios from the Dirichlet distribution $\text{Dir}(\alpha)$, where parameters $\alpha = 0.3$ and $\alpha = 0.6$ are used for this purpose.For more details about Dataset please refer to the **Appendix** A.

**Baselines.** The baselines used for comparison include several state-of-the-art (SOTA) methods in both the CFL and DFL settings. In the CFL, the baselines consist of FedAvg (McMahan et al., 2017). For the symmetric communication topology, the baselines include D-PSGD (Lian et al., 2017), DFedAvg, DFedAvgM (Sun et al., 2022), and DFedSAM (Shi et al., 2023). For the asymmetric communication topology, the baselines include SGP (Assran et al., 2019) and OSGP (Assran et al., 2019). These baselines are selected to provide comprehensive comparisons across different settings and methodologies. For more details about baselines please refer to the **Appendix** A.

**Implementation Details.** The total number of clients is set to 100, with 10% of the clients participating in the communication. For decentralized methods, all clients perform the local iteration step, while for centralized methods, only the participating clients perform the local update (Shi et al., 2023). The local learning rate is initialized to 0.1 with a decay rate of 0.998 per communication round for all experiments. For SAM-based algorithms, we set the perturbation weight as $\rho = 0.25$ for symmetric topology and $\rho = 0.1$ for asymetric topology. For more details about implementation and communication configurations please refer to the **Appendix** A.

### 5.2 PERFORMANCE EVALUATION

Table 1: Top 1 test accuracy (%) on three datasets in both IID and non-IID settings.

| Algorithm | MNIST | | | CIFAR-10 | | | CIFAR-100 | | |
|---|---|---|---|---|---|---|---|---|---|
| | Dir 0.3 | Dirt 0.6 | IID | Dir 0.3 | Dir 0.6 | IID | Dirt 0.3 | Dir 0.6 | IID |
| FedAvg | 97.33 | 98.10 | 98.29 | 78.98 | 80.02 | 81.34 | 55.20 | 56.41 | 57.35 |
| D-PSGD | 94.36 | 94.71 | 94.72 | 59.81 | 60.16 | 63.05 | 55.78 | 56.78 | 57.72 |
| DFedAvg | 97.77 | 97.82 | 98.08 | 77.37 | 77.94 | 80.05 | 58.27 | 58.72 | 59.31 |
| DFedAvgM | 98.14 | 98.29 | 98.30 | 79.54 | 80.73 | 82.92 | 58.06 | 58.52 | 59.37 |
| DFedSAM | 98.24 | 98.30 | 98.44 | 79.44 | 80.56 | 82.23 | 57.87 | 58.73 | 59.77 |
| SGP | 94.55 | 94.91 | 95.14 | 61.27 | 62.06 | 63.18 | 56.33 | 57.19 | 58.82 |
| OSGP | 97.85 | 97.95 | 98.08 | 76.59 | 77.93 | 79.97 | 58.50 | 58.87 | 59.34 |
| DFedSGPSM | **98.51** | 98.50 | **98.67** | 81.31 | 82.34 | **84.45** | 58.76 | **60.60** | 61.96 |
| DFedSGPSM-S | 98.43 | **98.53** | 98.53 | **81.35** | **82.36** | 83.84 | **58.80** | 60.21 | **62.10** |

**Comparative performance analysis with baseline methods.** In Table 1, we conducted a series of experiments on the MNIST, CIFAR-10, and CIFAR-100 datasets to compare our proposed algorithms, DFedSGPSM and DFedSGPSM-S, with all baselines from both the CFL and DFL settings. From Table 1 and Figure 1, it is evident that our proposed algorithms exhibit superior performance compared to other methods on these three datasets, regardless of whether in the DFL or CFL setting. To provide a more specific comparison, taking Dir-0.6 as an example, our algorithm outperforms

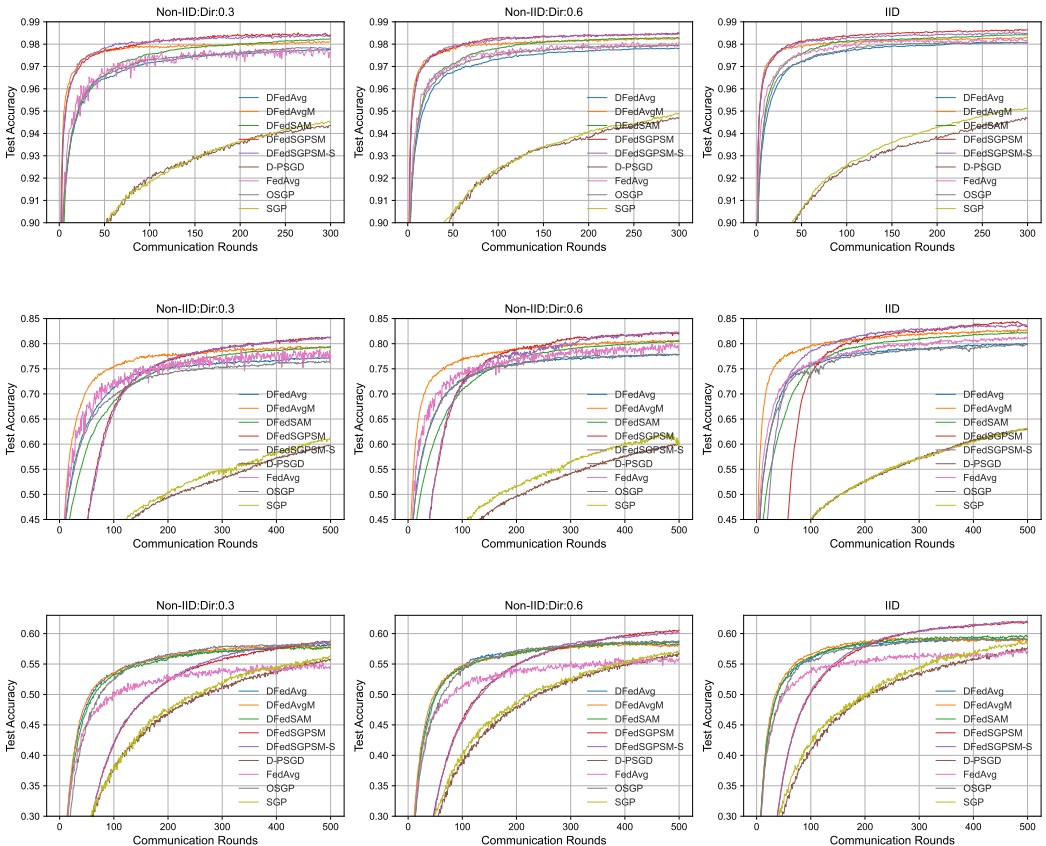

Figure 1: Test accuracy of all baselines on MNIST (first row), CIFAR-10 (second row), and CIFAR-100 (third row) in both IID and non-IID settings.

symmetric DFL methods with an accuracy lead of 0.27% in MNIST, 1.80% in CIFAR10, and 1.87% in CIFAR100. It is noteworthy that, under the same local optimizer, asymmetric DFL methods generally outperform symmetric DFL methods.

**Impact of non-IID levels($\alpha$).** From Figure 1 and Table 1, we can observe the robustness of our proposed algorithm in various non-IID scenarios. A smaller value of alpha ($\alpha$) indicates a higher level of non-IID, and correspondingly, the algorithm's task of optimizing a consensus problem becomes more challenging. Nevertheless, our algorithm still outperforms all baselines across different levels of non-IID. Moreover, methods based on the SAM optimizer consistently achieve higher accuracy across different levels of non-IID.

**Algorithm incorporating a neighbor selection strategy.** The neighbor selection strategy is a distinctive feature of asymmetric DFL compared to symmetric DFL. In asymmetric DFL, each client has the flexibility to choose $N_i^{out}$ based on certain specific rules. From Table 1 and Figure 1, we can observe that algorithms with a neighbor selection strategy do not significantly differ from those where each client randomly selects $N_i^{out}$ in terms of accuracy and convergence speed. However, it provides clients with a more flexible choice, allowing them to select $N_i^{out}$ according to certain rules or preferences, whereas in symmetric DFL, clients have almost no opportunity to choose $N_i^{out}$.

## 5.3 ABLATION STUDY

**Momentum Coefficient $\alpha$.** The convergence curves under different momentum coefficients after 500 communication rounds are displayed in Figure 2 (a) on the CIFAR10 dataset with Dir-0.3 partitioning. Here, $\alpha$ values are chosen from the set $\{0.1, 0.3, 0.5, 0.7, 0.9\}$. It is observed that as the value of $\alpha$ increases, the convergence speed of the algorithm gradually improves. However, for small or large values of $\alpha$, some degree of oscillation occurs in the later stages of convergence. Based on

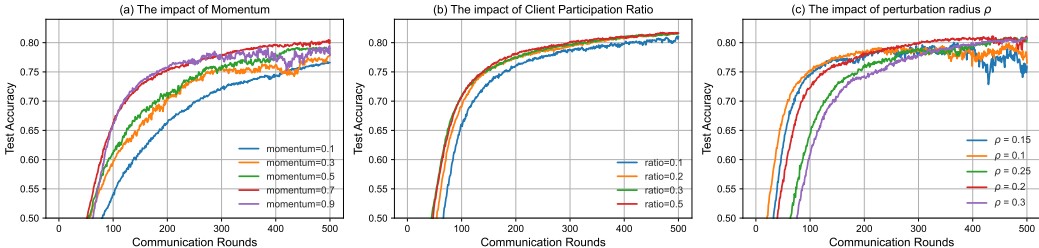

Figure 2: Sensitivity of hyper-parameters: momentum coefficient $\alpha$, participation ratio, perturbation radius $\rho$, respectively.

the results obtained from Figure 2 (a), the algorithm achieves the best generalization performance when $\alpha = 0.7$.

**Participation Ratio.** We conducted tests on the convergence performance of the algorithm under different client participation ratios. As depicted in Figure 2 (b), the highest generalization performance is achieved when the ratio is set to 0.5 from the set $\{0.1, 0.2, 0.3, 0.5\}$. Moreover, as the client participation ratio increases, the algorithm achieves faster convergence speed and better generalization performance. This is attributed to the fact that a higher client participation ratio enables each client to communicate with more neighbors in each communication round, but it also increases the communication burden. It is worth noting that our proposed algorithm is capable of achieving stable convergence even when the participation ratio is set to 0.1.

**Perturbation Radius $\rho$.** The perturbation radius $\rho$ is also one of the factors that influence the convergence of the algorithm. It controls the magnitude of perturbation in the SAM optimizer, and the impact of this perturbation accumulates as the number of communication rounds $T$ increases. We tested the generalization performance of the algorithm with different values of $\rho$ from the set $\{0.1, 0.15, 0.2, 0.25, 0.3\}$. Figure 2 (c) displays the highest accuracy achieved when $\rho = 0.2$. Additionally, when the value of $\rho$ is relatively small, our proposed algorithm exhibits some oscillations.

**Module Augmentation Ablation Experiments.** We conduct experiments on the CI-FAR10 dataset, using Dir-0.3 as the partitioning strategy. We incrementally add the Momentum, SAM, and Neighbor Selection modules to test the improvement in accuracy offered by each module. The results are presented in Table 2. We incorporate local momentum into the OSGP algorithm, resulting in a 2.13% increase in accuracy. Moreover, by introducing the SAM optimizer on top of the momentum inclusion, we achieve an additional accuracy improvement of 4.72%. Finally, shifting from random to the selection strategy mentioned in **Appendix** A.1 for each client's communication manner brings a 4.76% accuracy enhancement compared to OSGP. From the observed algorithm performance, it is apparent that the combination of Momentum and SAM significantly boosts the model's generalization ability. This is facilitated by SAM's capacity to generate locally flat models and search for models with uniformly low loss values, while Momentum accelerates the SAM's exploration for locally flat models.

Table 2: Comparison of Test Accuracy(%) for Different Algorithms.

| Algorithm | Momentum | SAM | Selection | Dir-0.3 |
|---|---|---|---|---|
| OSGP | | | | 76.59 |
| DFedSGPM | ✓ | | | 78.72 |
| DFedSGPSM | ✓ | ✓ | | 81.31 |
| DFedSGPSM-S | ✓ | ✓ | ✓ | 81.35 |

## 6 CONCLUSION.

In this paper, we propose a novel algorithm framework based on asymmetric DFL that combines the Push-Sum algorithm to achieve consensus optimization on a directed graph. To demonstrate the flexibility of client neighbor selection in a directed graph, our algorithm supports personalized neighbor selection strategies, allowing clients to choose which neighbors to send updates to. To further enhance the performance of the algorithm, we introduce momentum and the SAM optimizer, enabling clients to find local-flat models faster during optimization and avoid local overfitting. Theoretical analysis shows that our proposed algorithm achieves a convergence rate of $\mathcal{O}(\frac{1}{\sqrt{T}})$ in the non-convex setting. Empirically, we conducted extensive experiments on the MNIST, CIFAR10, and CIFAR100 datasets, demonstrating that the algorithm achieves state-of-the-art performance.

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

## A    MORE DETAILS ON ALGORITHM IMPLEMENTATION

**Datasets and backbones.** Three datasets are utilized in this study: MNIST, CIFAR10, and CIFAR100. MNIST is a relatively simple dataset for handwritten digit recognition, while CIFAR-10 and CIFAR-100 Krizhevsky et al. (2009) are labeled subsets of the larger 80 million images dataset. Each dataset comprises 50,000 images for training and 10,000 images for testing. In terms of the number of classes, both MNIST and CIFAR10 have 10 classes, while CIFAR100 has 100 classes. For the backbone architectures, ResNet-18 is employed for the more complex datasets. However, the batch-norm layers of ResNet-18 are replaced by group-norm layers due to the detrimental effect of batch-norm. For MNIST, the mnist_2NN architecture (Sun et al., 2022) is employed, which consists of two layers with 200 neurons each, followed by a fully connected layer for 10-class classification. As for CIFAR10, the network architecture includes two convolutional layers, one max pooling layer, and three fully connected layers. The first convolutional layer takes a 3-channel input and produces a 64-channel output using a kernel size of 5. Similarly, the second convolutional layer has 64 input channels, 64 output channels, and a kernel size of 5. Max pooling is performed after each convolutional layer with a kernel size of 2 and a stride of 2 to reduce spatial dimensions. The subsequent fully connected layers consist of 384 neurons in the first layer, 192 neurons in the second layer, and a final layer with the output size matching the number of classes in the classification task.

**More details about baselines.** FedAvg (McMahan et al., 2017) is a well-known classical FL method that trains a global model through weighted averaging in parallel with a central server. In the decentralized setting based on undirected graphs, D-PSGD (Lian et al., 2017) is a widely-used decentralized parallel SGD method that aims to achieve a consensus model. It is important to note that this study primarily focuses on decentralized FL, which involves multiple local iterations during training, as opposed to decentralized learning/training that is centered around one-step local training. For example, D-PSGD is a decentralized training algorithm that employs one-step SGD to train local models in each communication round. DFedAvg is a decentralized variant of FedAvg, where clients conduct local training and exchange updated models in communication rounds. DFedAvgM (Sun et al., 2022) extends DFedAvg by incorporating SGD with momentum, allowing each client to perform multiple local training steps before each communication round. Furthermore, DFedSAM (Shi et al., 2023) boosts the generalization capability of the aggregated model by applying gradient perturbation techniques to generate locally flattened models. In the realm of DFL based on directed graphs, SGP is a distributed algorithm that utilizes the push-sum algorithm to achieve consensus optimization on directed graphs. OSGP (Assran et al., 2019) builds upon SGP by supporting local multiple-round iterations before communication.

**Hyperparameters.** In our experimental setup, we employed a total of 100 clients for both the centralized and decentralized settings. Within the decentralized setting, each client is connected to a maximum of 10 neighbors. As for the centralized setting, the client sampling ratio is set to 0.1. The local learning rate is initialized to 0.1 and decays by a factor of 0.998 after every communication round across all experiments. In the case of centralized methods, the global learning rate is set to 1.0. The batch size remains constant at 128 for all experiments. We conducted 500 global communication rounds for the CIFAR-10/100 datasets and 300 global communication rounds for MNIST. For methods incorporating momentum, such as DFedAvgM and DFedSGPSM, we utilized a momentum coefficient of 0.9 for the MNIST and CIFAR-10 datasets, and 0.1 for the CIFAR-100 dataset. Additionally, for distributed methods like D-PSGD and SGP, we set the local epochs to 1, while other methods were set to 5.

**Communication configurations.** To ensure a fair comparison between decentralized and centralized approaches, we have incorporated a dynamic and time-varying connection topology for the decentralized methods. This ensures that the number of connections in each round does not exceed the number of connections in the centralized server. By controlling the number of client-to-client communications, we can match the communication volume of the centralized methods. It is important to mention that, following prior research, the communication complexity is evaluated in terms of the frequency of local communications. For the symmetric DFL, we generate a symmetric adjacency matrix where the number of neighbors each client has is determined by the client participation rate and the total number of clients. For the asymmetric DFL, we generate a column-random matrix which does not guarantee symmetry. Each client $i$ can send model parameters to other clients corresponding to non-zero weights in the $i$-th column of the column-random matrix. Additionally, (Dai et al., 2022b) pointed out that the communication overhead in DFL is greater than that in CFL. This

is because in DFL, each client needs to send model parameters to its neighbors, whereas in CFL, only the clients selected by the server need to send their parameters. To ensure a fair comparison, we control the number of neighbors for each client in DFL using the client participation rate. In this study, we set each client to have 10 neighbors.

**Implementation Details.** The number of communication rounds is set to 500 for all experiments comparing baselines on CIFAR-10/100, and 300 rounds for MNIST. Additionally, all ablation studies are conducted on the CIFAR-10 dataset with a data partition method of Dir-0.3 and 500 communication rounds. Additionally, we adopt the strategy mentioned in Section A.1 as the neighbor selection strategy for our proposed algorithm. We denote the algorithm with the neighbor selection strategy as DFedSGPSM-S.

## A.1 NEIGHBOR SELECTION

In order to highlight the flexibility of the communication topology of the directed graph, each client in the DFL system can flexibly choose its $N_i^{out}$. For this purpose, we have developed a neighbor selection strategy. For ease of exposition, we introduce a symbol $b_{i,j}$, which denotes $j \in N_i^{\text{out}}$. The probability that client $i$ selects client $j$ as a neighbor is:

$$p(b_{i,j}) = \frac{e^{|f_i - f_j|}}{\sum_{j=1}^{n} e^{|f_i - f_j|}} \tag{2}$$

Where $f_i$ represents the loss function value of client $i$, Equation 2 implies that if the difference between $f_i$ and $f_j$ is large, the probability of client $i$ selecting client $j$ as a neighbor will increase. This neighbor selection strategy has two advantages: (1) strengthening consistency among clients: by incorporating the model parameters corresponding to divergent $f_i$ and $f_j$, the differences between clients are continuously reduced, thereby enhancing consistency among clients; (2) enhancing convergence stability: as the neighbor selection is purposeful, it reduces the instability in convergence caused by random selection. It is worth noting that in Equation 2, it is assumed that client $i$ can receive $f$ values from all clients. However, in practice, client $i$ can only receive $f_j$ values from its neighbors $j \in N_i^{\text{in}}$. To address this limitation, the popular distributed consensus algorithm called the RAFT (Ongaro & Ousterhout, 2014) can be utilized, which enables client $i$ to obtain $f$ values from all clients. Please refer to (Ongaro & Ousterhout, 2014) for more details about the RAFT algorithm. In addition, we denote the algorithm with the neighbor selection strategy as DFedSGPSM-S.

## B APPENDIX

The Algorithm 1 can be expressed in a more concise form for all $t \geq 0$ and all $i = 1, \ldots, n$.

$$
\begin{aligned}
\mathbf{x}_i^{t+1} &= \sum_{j=1}^{n} p_{i,j}^t \mathbf{x}_j^{t+\frac{1}{2}} = \sum_{j \in N_i^{\text{in}}(t)} \mathbf{x}_j^{t+\frac{1}{2}} / d_j^t, \\
w_i^{t+1} &= \sum_{j=1}^{n} p_{i,j}^t w_j^t = \sum_{j \in N_i^{\text{in}}(t)} w_j^t / d_j^t, \\
\mathbf{z}_i^{t+1} &= \mathbf{x}_i^{t+1} / w_i^{t+1}, \\
\mathbf{x}_i^{t+\frac{1}{2}} &= \mathbf{x}_i^t - \eta_l^t \sum_{k=1}^{K} \sum_{s=1}^{k} \alpha^{k-s} \mathbf{g}_{i,s}^t
\end{aligned}
\tag{3}
$$

**Lemma 1** *Assume that the sequence $\{\mathbf{x}_i^t\}_{t\geq 0}$ is generated by Algorithm 1. for $\forall t \in \{1, 2, \cdots, T\}$ and $i \in \{1, 2, \cdots, n\}$, we have*

$$\mathbf{x}_i^{t+\frac{1}{2}} = \mathbf{x}_i^t - \eta_l \sum_{k=1}^{K} \sum_{s=1}^{k} \alpha^{k-s} \mathbf{g}_{i,s}^t \tag{4}$$

**Proof 1** *According to the update rule of Line.8 and Line.9 in Algorithm 1, we have:*

$$\mathbf{x}_{i,k+1}^t = \mathbf{x}_{i,k}^t - \eta_l \mathbf{g}_{i,k+1}^t + \alpha(\mathbf{x}_{i,k}^t - \mathbf{x}_{i,k-1}^t) \tag{5}$$

*Where $\mathbf{x}_{i,0}^t = \mathbf{x}_{i,-1}^t = 0$. Next, by defining $\Delta_{i,k+1}^t = \mathbf{x}_{i,k+1}^t - \mathbf{x}_{i,k}^t$, where $a_{i,0}^t = 0$, equation (5) can be transformed into the following form:*

$$\Delta_{i,k}^t = \alpha \Delta_{i,k-1}^t - \eta_l^t \mathbf{g}_{i,k}^t = \alpha^k \Delta_{i,0}^t - \eta_l \sum_{s=1}^{k} \alpha^{k-s} \mathbf{g}_{i,s}^t = -\eta_l \sum_{s=1}^{k} \alpha^{k-s} \mathbf{g}_{i,s}^t$$

*By summing up the iteration $k$ from 1 to K, we obtain the following equation:*

$$\mathbf{x}_{i,K}^t - \mathbf{x}_{i,0}^t = \sum_{k=1}^{K} \Delta_{i,k}^t = -\eta_l \sum_{k=1}^{K} \sum_{s=1}^{k} \alpha^{k-s} \mathbf{g}_{i,s}^t$$

*Since $\mathbf{x}_i^{t+\frac{1}{2}} = \mathbf{x}_{i,K}^t$ in line 11 of Algorithm 1 and $\mathbf{x}_{i,0}^t = \mathbf{x}_i^t$, we have completed the proof.*

**Lemma 2** *[Sun et al. (2022) Lemma 2] Assume that Assumption 4, 5 hold, and $0 < \alpha < 1$. Let $(\mathbf{x}_{i,k}^t)_{t \geq 0}$ be generated by the Algorithm 1, It then follows*

$$\mathbb{E}\|\mathbf{x}_{i,k+1}^t - \mathbf{x}_{i,k}^t\|^2 \leq \frac{1}{(1-\alpha)^2}(2\eta_l^2 \sigma_l^2 + 2\eta_l^2 B^2)$$

*When $0 \leq k \leq K-1$.*

***Proof 2*** *For detailed proof, please refer to the reference by Sun et al. (2022).*

**Lemma 3** *[Sun et al. (2022) Lemma 3] Given the stepsize $0 < \eta_l < \frac{\delta}{8LK}$ and $i \in \{1, 2, \cdots, n\}$ and $(\mathbf{x}_{i,k}^t)_{t,k \geq 0}$, $(\mathbf{x}_i^t)_{t \geq 0}$ are generated by the Algorithm 1 for all $i \in \{1, 2, \cdots, n\}$. If Assumption 5 holds, it then follows*

$$\frac{1}{n} \sum_{i=1}^{n} \mathbb{E}\|\mathbf{x}_{i,k}^t - \mathbf{x}_i^t\|^2 \leq C_1 \eta_l^2 + 64K^2 \eta_l^2 \frac{\sum_{i=1}^{n} \mathbb{E}\|\nabla f(\mathbf{z}_i^t)\|^2}{n}$$

*Where $C_1 := 8K\sigma_l^2 + 64K^2\sigma_g^2 + \frac{32K^2\alpha^2}{(1-\alpha)^2}(\sigma_l^2 + B^2) + 64KL^2\rho^2$ when $0 \leq k \leq K$. and $\delta > 0$ is a constant and its definition can be found in Proposition 2.1 by Taheri et al. (2020).*

***Proof 3*** *According to the equation (5), we have*

$$\mathbb{E}\|\mathbf{x}_{i,k+1}^t - \mathbf{x}_i^t\|^2 = \mathbb{E}\|\mathbf{x}_{i,k}^t - \eta_l \mathbf{g}_{i,k}^t - \mathbf{x}_i^t + \alpha(\mathbf{x}_{i,k}^t - \mathbf{x}_{i,k-1}^t)\|^2$$

$$\leq (1 + \frac{1}{2K-1})\mathbb{E}\|\mathbf{x}_{i,k}^t - \mathbf{x}_i^t\|^2 + 2K\mathbb{E}\| - \eta_l \mathbf{g}_{i,k}^t + \alpha(\mathbf{x}_{i,k}^t - \mathbf{x}_{i,k-1}^t)\|^2$$

$$\leq (1 + \frac{1}{2K-1})\mathbb{E}\|\mathbf{x}_{i,k}^t - \mathbf{x}_i^t\|^2 + 2K\eta_l^2\sigma_l^2 + 2K\mathbb{E}\| - \eta_l \nabla f_i(\check{\mathbf{z}}_{i,k}^t) + \alpha(\mathbf{x}_{i,k}^t - \mathbf{x}_{i,k-1}^t)\|^2$$

$$\leq (1 + \frac{1}{2K-1})\mathbb{E}\|\mathbf{x}_{i,k}^t - \mathbf{x}_i^t\|^2 + 2K\eta_l^2\sigma_l^2 + 4K\eta_l^2\mathbb{E}\|\nabla f_i(\check{\mathbf{z}}_{i,k}^t)\|^2 + 4K\alpha^2\mathbb{E}\|(\mathbf{x}_{i,k}^t - \mathbf{x}_{i,k-1}^t)\|^2$$

$$\leq (1 + \frac{1}{2K-1})\mathbb{E}\|\mathbf{x}_{i,k}^t - \mathbf{x}_i^t\|^2 + 2K\eta_l^2\sigma_l^2 + 4K\eta_l^2\mathbb{E}\|\nabla f_i(\check{\mathbf{z}}_{i,k}^t)\|^2 + \frac{4K\alpha^2}{(1-\alpha)^2}(2\eta_l^2\sigma_l^2 + 2\eta_l^2 B^2)$$

*where the last inequality is derived from Lemma 2. Next we will bound the term $\mathbb{E}\|\nabla f_i(\check{\mathbf{z}}_{i,k}^t)\|^2$.*

$$\mathbb{E}\|\nabla f_i(\check{\mathbf{z}}_{i,k}^t)\|^2 = \mathbb{E}\|\nabla f_i(\check{\mathbf{z}}_{i,k}^t) - \nabla f_i(\mathbf{z}_{i,k}^t) + \nabla f_i(\mathbf{z}_{i,k}^t) - \nabla f(\mathbf{z}_{i,k}^t) + \nabla f(\mathbf{z}_{i,k}^t) - \nabla f(\mathbf{z}_i^t) + \nabla f(\mathbf{z}_i^t)\|^2$$

$$\leq 4L^2\rho^2 + 4\mathbb{E}\|\nabla f_i(\mathbf{z}_{i,k}^t) - \nabla f(\mathbf{z}_{i,k}^t)\|^2 + 4\mathbb{E}\|\nabla f(\mathbf{z}_{i,k}^t) - \nabla f(\mathbf{z}_i^t)\|^2 + 4\mathbb{E}\|\nabla f(\mathbf{z}_i^t)\|^2$$

$$\leq 4L^2\rho^2 + 4\sigma_g^2 + 4L^2\mathbb{E}\|\mathbf{z}_{i,k}^t - \mathbf{z}_i^t\|^2 + 4\mathbb{E}\|\nabla f(\mathbf{z}_i^t)\|^2$$

*where the last inequality is based on assumptions 2 and 3. In addition, according to line 5 of Algorithm 1, we can obtain $\mathbb{E}\|\mathbf{z}_{i,k}^t - \mathbf{z}_i^t\|^2 = \frac{1}{\|w_i^t\|^2}\mathbb{E}\|\mathbf{x}_{i,k}^t - \mathbf{x}_i^t\|^2$. According to Property 2.1 by Taheri et al. (2020), We obtain that there exists $\delta > 0$ such that $\|w_i^t\| > \delta$. Therefore, we have $\mathbb{E}\|\mathbf{z}_{i,k}^t - \mathbf{z}_i^t\|^2 \leq \frac{1}{\delta^2}\mathbb{E}\|\mathbf{x}_{i,k}^t - \mathbf{x}_i^t\|^2$. Then we have*

$$\mathbb{E}\|\nabla f_i(\check{\mathbf{z}}_{i,k}^t)\|^2 \leq 4L^2\rho^2 + 4\sigma_g^2 + 4\frac{L^2}{\delta^2}\mathbb{E}\|\mathbf{x}_{i,k}^t - \mathbf{x}_i^t\|^2 + 4\mathbb{E}\|\nabla f(\mathbf{z}_i^t)\|^2$$

*Then, we get*

$$
\begin{aligned}
\mathbb{E}\|\mathbf{x}_{i,k+1}^t - \mathbf{x}_i^t\|^2 &\leq (1 + \frac{1}{2K-1} + \frac{16KL^2\eta_l^2}{\delta^2})\mathbb{E}\|\mathbf{x}_{i,k}^t - \mathbf{x}_i^t\|^2 + 2K\eta_l^2\sigma_l^2 + 16K\eta_l^2\sigma_g^2 \\
&\quad + 16K\eta_l^2\mathbb{E}\|\nabla f(\mathbf{z}_i^t)\|^2 + \frac{4K\alpha^2}{(1-\alpha)^2}(2\eta_l^2\sigma_l^2 + 2\eta_l^2 B^2) + 16KL^2\rho^2\eta_l^2 \\
&\leq (1 + \frac{1}{K-1})\mathbb{E}\|\mathbf{x}_{i,k}^t - \mathbf{x}_i^t\|^2 + 2K\eta_l^2\sigma_l^2 + 16K\eta_l^2\sigma_g^2 + \frac{4K\alpha^2}{(1-\alpha)^2}(2\eta_l^2\sigma_l^2 + 2\eta_l^2 B^2) \\
&\quad + 16K\eta_l^2\mathbb{E}\|\nabla f(\mathbf{z}_i^t)\|^2 + 16KL^2\rho^2\eta_l^2 \\
&\leq \sum_{j=0}^{K-1}(1 + \frac{1}{K-1})^j [2K\eta_l^2\sigma_l^2 + 16K\eta_l^2\sigma_g^2 + 16K\eta_l^2\mathbb{E}\|\nabla f(\mathbf{z}_i^t)\|^2 + \frac{4K\alpha^2}{(1-\alpha)^2}(2\eta_l^2\sigma_l^2 + 2\eta_l^2 B^2) \\
&\quad + 16KL^2\rho^2\eta_l^2] \\
&\leq (K-1)[(1 + \frac{1}{K-1})^K - 1] \times [2K\eta_l^2\sigma_l^2 + 16K\eta_l^2\sigma_g^2 + \frac{4K\alpha^2}{(1-\alpha)^2}(2\eta_l^2\sigma_l^2 + 2\eta_l^2 B^2) \\
&\quad + 16K\eta_l^2\mathbb{E}\|\nabla f(\mathbf{z}_i^t)\|^2 + 16KL^2\rho^2\eta_l^2] \\
&\leq 8K^2\eta_l^2\sigma_l^2 + 64K^2\eta_l^2\sigma_g^2 + \frac{32K^2\alpha^2}{(1-\alpha)^2}(\eta_l^2\sigma_l^2 + \eta_l^2 B^2) + 64K^2\eta_l^2\mathbb{E}\|\nabla f(\mathbf{z}_i^t)\|^2 + 64KL^2\rho^2\eta_l^2
\end{aligned}
$$

*Where we used the inequality $(1 + \frac{1}{K-1})^K \leq 5$ holds for any $K > 1$. Thus we complete the proof.*

**Lemma 4** *Suppose the time-varying communication topology is strongly connected. It holds for $\forall i \in \{1, \cdots, n\}$ and $t \geq 0$ that*

$$
\mathbb{E}\|\mathbf{z}_i^t - \bar{\mathbf{x}}^t\|^2 \leq 2C^2 q^{2t}\|\mathbf{x}_i^0\|^2 + 2\eta_l^2\tilde{\alpha}^2 C^2 \frac{B^2 + \sigma_l^2}{(1-q)^2}
$$

*Where the constants $q \in (0,1)$ and $C > 0$ are related to the maximum number of out-neighbor $|\mathcal{N}_{max}^{out}| = \max\{|\mathcal{N}_i^{out}|, i \in \{1, \cdots, n\}\}$ and $\mathcal{B}$, $\bar{\mathbf{x}}^t$ denotes the average of local model at global iteration $t$, $\tilde{\alpha} = \sum_{k=1}^{K}\sum_{s=1}^{k}\alpha^{k-s}$, it is evident that $0 < \tilde{\alpha} < \frac{K}{1-\alpha}$.*

**Proof 4** *Based on Lemma 3 in Assran et al. (2019), we can obtain the following expression:*

$$
\|\mathbf{z}_i^t - \bar{\mathbf{x}}^t\| \leq Cq^t\|\mathbf{x}_i^0\| + C\sum_{j=0}^{t} q^{t-j}\|\eta_l \sum_{k=1}^{K}\sum_{s=1}^{k}\alpha^{k-s}\mathbf{g}_{i,s}^j\| \tag{6}
$$

*Based on Assumption 5, 3 and setting $\tilde{\alpha} = \sum_{k=1}^{K}\sum_{s=1}^{k}\alpha^{k-s}$. We can now obtain the following expression:*

$$
\|\eta_l \sum_{k=1}^{K}\sum_{s=1}^{k}\alpha^{k-s}\mathbf{g}_{i,s}^j\| \leq \eta_l\tilde{\alpha}\sqrt{B^2 + \sigma_l^2}
$$

*Then the inequality (6) can be simplified to the following expression:*

$$
\|\mathbf{z}_i^t - \bar{\mathbf{x}}^t\| \leq Cq^t\|\mathbf{x}_i^0\| + \eta_l\tilde{\alpha}C\frac{\sqrt{B^2 + \sigma_l^2}}{1-q}
$$

*After squaring both sides and taking expectations, we have completed the proof.*

**Lemma 5** *Let Assumption 2-5 hold and $C > 0, q \in (0,1)$ are the two non-negative constants defined in Lemma 4, Then*

**Proof 5** *According to the Algorithm 1 line 13, we get*

$$
f(\bar{\mathbf{x}}^{t+1}) = f(\frac{1}{n}\sum_{i=1}^{n}\mathbf{x}_i^{t+1}) = f(\frac{1}{n}\sum_{i=1}^{n}\sum_{j=1}^{n}p_{i,j}^t\mathbf{x}_j^{t+\frac{1}{2}})
$$

*Since the communication topology is a column-stochastic matrix, therefore*

$$\sum_{i=1}^{n}\sum_{j=1}^{n}p_{i,j}^{t}\mathbf{x}_{j}^{t+\frac{1}{2}} = \sum_{j=1}^{n}\sum_{i=1}^{n}p_{i,j}^{t}\mathbf{x}_{j}^{t+\frac{1}{2}} = \sum_{j=1}^{n}\mathbf{x}_{j}^{t+\frac{1}{2}} = \sum_{i=1}^{n}\mathbf{x}_{i}^{t+\frac{1}{2}}$$

*Combining the above two equations and using Lemma 3, we can obtain*

$$f(\bar{\mathbf{x}}^{t+1}) = f(\frac{1}{n}\sum_{i=1}^{n}\mathbf{x}_{j}^{t+\frac{1}{2}}) = f(\frac{1}{n}\sum_{i=1}^{n}(\mathbf{x}_{i}^{t} - \eta_{l}\sum_{k=1}^{K}\sum_{s=1}^{k}\alpha^{k-s}\mathbf{g}_{i,s}^{t}))$$

$$= f(\bar{\mathbf{x}}^{t} - \eta_{l}\frac{1}{n}\sum_{i=1}^{n}\sum_{k=1}^{K}\sum_{s=1}^{k}\alpha^{k-s}\mathbf{g}_{i,s}^{t})$$

$$\leq f(\bar{\mathbf{x}}^{t}) - \eta_{l}\langle\nabla f(\bar{\mathbf{x}}^{t}), \frac{1}{n}\sum_{i=1}^{n}\sum_{k=1}^{K}\sum_{s=1}^{k}\alpha^{k-s}\mathbf{g}_{i,s}^{t}\rangle + \frac{L\eta_{l}^{2}}{2}\|\frac{1}{n}\sum_{i=1}^{n}\sum_{k=1}^{K}\sum_{s=1}^{k}\alpha^{k-s}\mathbf{g}_{i,s}^{t}\|^{2}$$

*Taking conditional expectations of both sides simultaneously and let $\tilde{\alpha} = \sum_{k=1}^{K}\sum_{s=1}^{k}\alpha^{k-s}$, we have*

$$\mathbb{E}f(\bar{\mathbf{x}}^{t+1}) \leq \mathbb{E}f(\bar{\mathbf{x}}^{t}) - \eta_{l}\mathbb{E}\langle\nabla f(\bar{\mathbf{x}}^{t}), \frac{1}{n}\sum_{i=1}^{n}\sum_{k=1}^{K}\sum_{s=1}^{k}\alpha^{k-s}\nabla f_{i}(\check{\mathbf{z}}_{i,s}^{t})\rangle + \frac{L\eta_{l}^{2}}{2}\mathbb{E}\|\frac{1}{n}\sum_{i=1}^{n}\sum_{k=1}^{K}\sum_{s=1}^{k}\alpha^{k-s}\mathbf{g}_{i,s}^{t}\|^{2}$$

$$= \mathbb{E}f(\bar{\mathbf{x}}^{t}) - \eta_{l}\tilde{\alpha}\mathbb{E}\langle\nabla f(\bar{\mathbf{x}}^{t}), \frac{1}{n\tilde{\alpha}}\sum_{i=1}^{n}\sum_{k=1}^{K}\sum_{s=1}^{k}\alpha^{k-s}\nabla f_{i}(\check{\mathbf{z}}_{i,s}^{t})\rangle + \frac{L\eta_{l}^{2}}{2}\mathbb{E}\|\frac{1}{n}\sum_{i=1}^{n}\sum_{k=1}^{K}\sum_{s=1}^{k}\alpha^{k-s}\mathbf{g}_{i,s}^{t}\|^{2}$$

$$\leq \mathbb{E}f(\bar{\mathbf{x}}^{t}) - \frac{\tilde{\alpha}\eta_{l}}{2}\mathbb{E}\|\nabla f(\bar{\mathbf{x}}^{t})\|^{2} - \frac{\tilde{\alpha}\eta_{l}}{2}\mathbb{E}\|\frac{1}{n\tilde{\alpha}}\sum_{i=1}^{n}\sum_{k=1}^{K}\sum_{s=1}^{k}\alpha^{k-s}\nabla f_{i}(\check{\mathbf{z}}_{i,s}^{t})\|^{2} + \frac{L\eta_{l}^{2}\tilde{\alpha}^{2}\sigma_{l}^{2}}{2}$$

$$+ \frac{\tilde{\alpha}\eta_{l}}{2}\mathbb{E}\|\frac{1}{n\tilde{\alpha}}\sum_{i=1}^{n}\sum_{k=1}^{K}\sum_{s=1}^{k}\alpha^{k-s}\left(\nabla f_{i}(\check{\mathbf{z}}_{i,s}^{t}) - \nabla f(\bar{\mathbf{x}}^{t})\right)\|^{2} + \frac{L\eta_{l}^{2}\tilde{\alpha}^{2}}{2}\mathbb{E}\|\frac{1}{n\tilde{\alpha}}\sum_{i=1}^{n}\sum_{k=1}^{K}\sum_{s=1}^{k}\alpha^{k-s}\nabla f_{i}(\check{\mathbf{z}}_{i,s}^{t})\|^{2}$$

$$= \mathbb{E}f(\bar{\mathbf{x}}^{t}) - \frac{\tilde{\alpha}\eta_{l}}{2}\mathbb{E}\|\nabla f(\bar{\mathbf{x}}^{t})\|^{2} - \frac{\tilde{\alpha}\eta_{l} - L\eta_{l}^{2}\tilde{\alpha}^{2}}{2}\mathbb{E}\|\frac{1}{n\tilde{\alpha}}\sum_{i=1}^{n}\sum_{k=1}^{K}\sum_{s=1}^{k}\alpha^{k-s}\nabla f_{i}(\check{\mathbf{z}}_{i,s}^{t})\|^{2} + \frac{L\eta_{l}^{2}\tilde{\alpha}^{2}\sigma_{l}^{2}}{2}$$

$$+ \underbrace{\frac{\tilde{\alpha}\eta_{l}}{2}\mathbb{E}\|\frac{1}{n\tilde{\alpha}}\sum_{i=1}^{n}\sum_{k=1}^{K}\sum_{s=1}^{k}\alpha^{k-s}\left(\nabla f_{i}(\check{\mathbf{z}}_{i,s}^{t}) - \nabla f(\bar{\mathbf{x}}^{t})\right)\|^{2}}_{\textbf{R1}}$$

*Let us now focus on finding an upper bound for the quantity* **R1**

$$\mathbf{R1} \overset{Jensen}{\leq} \frac{1}{n\tilde{\alpha}} \sum_{i=1}^{n} \sum_{k=1}^{K} \sum_{s=1}^{k} \alpha^{k-s} \mathbb{E}\|\nabla f_i(\breve{\mathbf{z}}_{i,s}^t) - \nabla f(\bar{\mathbf{x}}^t)\|^2$$

$$\overset{Le2}{\leq} \frac{L^2}{n\tilde{\alpha}} \sum_{i=1}^{n} \sum_{k=1}^{K} \sum_{s=1}^{k} \alpha^{k-s} \mathbb{E}\|\breve{\mathbf{z}}_{i,s}^t - \mathbf{z}_{i,s}^t + \mathbf{z}_{i,s}^t - \mathbf{z}_i^t + \mathbf{z}_i^t - \bar{\mathbf{x}}^t\|^2$$

$$\leq \frac{3L^2}{n\tilde{\alpha}} \sum_{i=1}^{n} \sum_{k=1}^{K} \sum_{s=1}^{k} \alpha^{k-s} \mathbb{E}\|\mathbf{z}_{i,s}^t - \mathbf{z}_i^t\|^2 + \frac{3L^2}{n\tilde{\alpha}} \sum_{i=1}^{n} \sum_{k=1}^{K} \sum_{s=1}^{k} \alpha^{k-s} \mathbb{E}\|\mathbf{z}_i^t - \bar{\mathbf{x}}^t\|^2 + 3L^2\rho^2$$

$$\leq \frac{3L^2}{n\tilde{\alpha}\delta} \sum_{i=1}^{n} \sum_{k=1}^{K} \sum_{s=1}^{k} \alpha^{k-s} \mathbb{E}\|\mathbf{x}_{i,s}^t - \mathbf{x}_i^t\|^2 + \frac{3L^2}{n\tilde{\alpha}} \sum_{i=1}^{n} \sum_{k=1}^{K} \sum_{s=1}^{k} \alpha^{k-s} \mathbb{E}\|\mathbf{z}_i^t - \bar{\mathbf{x}}^t\|^2 + 3L^2\rho^2$$

$$\overset{Le3,4}{\leq} 6L^2 \left( C^2 q^{2t} \frac{1}{n} \sum_{i=1}^{n} \|\mathbf{x}_i^0\|^2 + \eta_l^2 \tilde{\alpha}^2 C^2 \frac{B^2 + \sigma_l^2}{(1-q)^2} \right) + 3L^2\rho^2$$

$$+ \frac{3L^2\eta_l^2}{\delta} \left( C_1 + 64K^2 \frac{\sum_{i=1}^{n} \mathbb{E}\|\nabla f(\mathbf{z}_i^t)\|^2}{n} \right)$$

$$\overset{Ass5}{\leq} \frac{3L^2\eta_l^2}{\delta} \left( C_1 + 64K^2 B^2 \right) + 6L^2 \left( C^2 q^{2t} \frac{1}{n} \sum_{i=1}^{n} \|\mathbf{x}_i^0\|^2 + \eta_l^2 \tilde{\alpha}^2 C^2 \frac{B^2 + \sigma_l^2}{(1-q)^2} \right) + 3L^2\rho^2$$

*Assuming that* $0 < \eta_l < \frac{1}{KL}$. *Then* $\tilde{\alpha}\eta_l - L\eta_l^2\tilde{\alpha}^2 > 0$. *Substituting* **R1** *into the expression, we have:*

$$\mathbb{E}f(\bar{\mathbf{x}}^{t+1}) \leq \mathbb{E}f(\bar{\mathbf{x}}^t) - \frac{\tilde{\alpha}\eta_l}{2} \mathbb{E}\|\nabla f(\bar{\mathbf{x}}^t)\|^2 + \frac{L\eta_l^2\tilde{\alpha}^2\sigma_l^2}{2} + \frac{3\tilde{\alpha}\eta_l^3 L^2}{2\delta} \left( C_1 + 64K^2 B^2 \right)$$

$$+ 3L^2 C^2 q^{2t} \eta_l \tilde{\alpha} \frac{1}{n} \sum_{i=1}^{n} \|\mathbf{x}_i^0\|^2 + 3L^2 \eta_l^3 \tilde{\alpha}^3 C^2 \frac{B^2 + \sigma_l^2}{(1-q)^2} + \frac{3}{2} \tilde{\alpha}\eta_l L^2 \rho^2$$

*By rearranging and dividing all terms by* $\frac{\tilde{\alpha}\eta_l}{2}$ *we obtain:*

$$\mathbb{E}\|\nabla f(\bar{\mathbf{x}}^t)\|^2 \leq \frac{2 \left( \mathbb{E}f(\bar{\mathbf{x}}^t) - \mathbb{E}f(\bar{\mathbf{x}}^{t+1}) \right)}{\tilde{\alpha}\eta_l} + L\eta_l\tilde{\alpha}\sigma_l^2 + \frac{3\eta_l^2 L^2}{\delta} \left( C_1 + 64K^2 B^2 \right)$$

$$+ 6L^2 C^2 q^{2t} \frac{1}{n} \sum_{i=1}^{n} \|\mathbf{x}_i^0\|^2 + 6L^2 \eta_l^2 \tilde{\alpha}^2 C^2 \frac{B^2 + \sigma_l^2}{(1-q)^2} + 3L^2\rho^2$$

*Taking the sum over the index* $t$ *and let* $f^*$ *be the minimum value of* $f(\mathbf{x})$, *we obtain:*

$$\frac{1}{T} \sum_{t=0}^{T-1} \mathbb{E}\|\nabla f(\bar{\mathbf{x}}^t)\|^2 \leq \frac{2 \left( f(\bar{\mathbf{x}}^0) - f^* \right)}{T\tilde{\alpha}\eta_l} + L\eta_l\tilde{\alpha}\sigma_l^2 + \frac{3\eta_l^2 L^2}{\delta} \left( C_1 + 64K^2 B^2 \right)$$

$$+ \frac{6L^2 C^2}{T(1-q^2)} \frac{1}{n} \sum_{i=1}^{n} \|\mathbf{x}_i^0\|^2 + 6L^2 \eta_l^2 \tilde{\alpha}^2 C^2 \frac{B^2 + \sigma_l^2}{(1-q)^2} + 3L^2\rho^2$$

*Finally, utilizing* $0 < \tilde{\alpha} < \frac{K}{1-\alpha}$, *we obtain the final result:*

$$\frac{1}{T} \sum_{t=0}^{T-1} \mathbb{E}\|\nabla f(\bar{\mathbf{x}}^t)\|^2 \leq \frac{2(1-\alpha) \left( f(\bar{\mathbf{x}}^0) - f^* \right)}{T\eta_l K} + \frac{L\eta_l K\sigma_l^2}{1-\alpha} + \frac{3\eta_l^2 L^2}{\delta} \left( C_1 + 64K^2 B^2 \right)$$

$$+ \frac{6L^2 C^2}{T(1-q^2)} \frac{1}{n} \sum_{i=1}^{n} \|\mathbf{x}_i^0\|^2 + 6L^2 \eta_l^2 K^2 C^2 \frac{(B^2 + \sigma_l^2)}{(1-\alpha)^2(1-q)^2} + 3L^2\rho^2$$

