# OpenReview forum: "Asymmetrically Decentralized Federated Learning"
_ICLR.cc/2024/Conference — ICLR 2024 Conference Withdrawn Submission_

### Official Review · Reviewer_wyFy · 2023-11-03

**Soundness:** 3 good
**Presentation:** 3 good
**Contribution:** 1 poor
**Rating:** 3
**Confidence:** 5

**Summary:**

This paper considers decentralzied federated learning problems over general directed communication networks. To solve this problem, the authors propose the DFedSGPSM algorithm, which utilizes the Push-Sum protocol to deal with the asymmetry of graphs, and employs the Sharpness Aware Minimization (SAM) optimizer and local momentum to mitigate local heterogeneous overfitting and improve algorithm performance. Their analysis shows that the proposed DFedSGPSM achieves a convergence rate of $ \mathcal{O}\left( \frac{1}{\sqrt{T}} \right) $ in a non-convex smooth setting. They conduct extensive experiments on three datasets, and demonstrate the superior performance of DFedSGPSM compared to the existing state-of-the-art algorithms.

**Strengths:**

1. They provide theoretical convergence results for their proposed DFedSGPSM and discuss the impacts of some important problem-related parameters such as $C$, $q$ and $L$ on the convergence rate.

2. They conduct extensive experiments, and the ablation study provides a detailed discussion in different settings.

3. The paper is generally well written and easy to follow.

**Weaknesses:**

1. The novlty of the proposed algorithm is limited. It seems that DFedSGPSM is a combination of DFedSAM [1] with Push-Sum protocol [2] and momentum mechanism, and the convergence analysis can be thus directly extended by combining existing theoretical analysis methods. The authors should comment on this and clarify what’s the new technical challenge in their convergence analysis.

2. According to Corollary 1, DFedSGPSM can converge at a rate of $\mathcal{O}(1/\sqrt{T})$. However, existing decentralized approaches such as SGP [2] and D-PSGD [3] can converge at a rate of $\mathcal{O}(1/\sqrt{nT}) $ where $n$ is the size of the network, which shows a clear linear speedup in the node number. The reviewer believes it is the major limitation in the convergence property of DFedSGPSM. Can the authors improve the rate, or comment on the reason why they cannot achieve the linear speedup ?

3. The authors assume that the gradient is bounded (Assumption 5), which is too restrictive.  DFedSAM [1], SGP [2], and D-PSGD [3] do not need this assumption. With this assumption, the theoretical analysis becomes quite straightfoward. In fact, the gradient can be bounded by $ \left\| \nabla f_i\left( x \right) \right\| ^2\leqslant 2\left\| \nabla f_i\left( x \right) -\nabla f\left( x \right) \right\| ^2+\left\| \nabla f\left( x \right) \right\| ^2\leqslant 2\sigma _{g}^{2}+\left\| \nabla f\left( x \right) \right\| ^2 $ .

4. The impact of momentum parameter $\alpha$ on the convergence rate is not discussed in their theeoretical results.

5. Empirically, they lack important experiments on different node scales, and the scalability of the algorithm with respect to the number of nodes is not clear.

[1] Yifan Shi, Li Shen, Kang Wei, Yan Sun, Bo Yuan, Xueqian Wang, and Dacheng Tao. Improving the model consistency of decentralized federated learning. arXiv preprint arXiv:2302.04083, 2023.

[2] Mahmoud Assran, Nicolas Loizou, Nicolas Ballas, and Mike Rabbat. Stochastic gradient push for distributed deep learning. In International Conference on Machine Learning, pp. 344–353. PMLR, 2019.

[3] Xiangru Lian, Ce Zhang, Huan Zhang, Cho-Jui Hsieh, Wei Zhang, and Ji Liu. Can decentralized algorithms outperform centralized algorithms? a case study for decentralized parallel stochastic gradient descent. In Advances in Neural Information Processing Systems, pp. 5330–5340, 2017.

**Questions:**

1. In their proof of Lemma 1 in the appendix, their statement “Where $x_{i,0}^{t}=x_{i,-1}^{t}=0$” seems not correct. The reviewer believe that it should be $ x_{i,0}^{t}=x_{i,-1}^{t}$. The authors should check this.

2. It seems that the line number of the algorithm they refer does not correspond to what they really used. For example, “Line. 8 and Line. 9” should be replaced by “Line. 10 and Line. 11” in the proof of Lemma 1 in the appendix. The authors should check this.

3. Should Assumption 4 be replaced by Assumption 3 in the Lemma 2 in the appendix ?

**Details Of Ethics Concerns:**

N.A.

---

### Official Review · Reviewer_38tq · 2023-11-04

**Soundness:** 2 fair
**Presentation:** 2 fair
**Contribution:** 2 fair
**Rating:** 5
**Confidence:** 4

**Summary:**

The paper combines and applies three existing methods for decentralized federated learning (DFL) under the asymmetric setting. The first method is using momentum. The second method is using the push-sum protocol. The third method is using sharpness aware minimization optimization. Convergence rate result is provided. Experiments are conducted on both iid and non-iid federated learning datasets.

**Strengths:**

1. The paper explores the area of decentralized federated learning (asymmetric communication) under controlled heterogeneity data setting (non-iid).
2. The paper is easy to follow.
3. The paper provides theoretical analysis (convergence result).
4. The paper provides ablation study as it relates to the three combined methods.

**Weaknesses:**

1. Novelty is on the weaker side/incremental since the paper combines existing methods. It is not surprising to see the combination of these methods improving performance.
2. Each client can choose $N_i^{out}$ in asymmetric DFL, but results show performance results do not significantly differ with rule-based or random $N_i^{out}$ selection. Flexibility is the advantage (over symmetric) given by the paper. However, this is an unsatisfactory reason if it is related to topological connectivity.
3. Source code not given so unclear if results are reproducible
4. Would be helpful to see $\pm$ information for test accuracy results in Table 1.

**Questions:**

Since the paper mentions DFedSAM (Shi et al., 2023) is a most relevant work, I was curious on the test accuracy results (curve) for DFedSAM shown in Figure 1 of this paper with 1000 communication rounds as a comparison to the results shown in Figure 3 of the DFedSAM paper (https://arxiv.org/pdf/2302.04083.pdf) for CIFAR-10 and CIFAR-100.

---

### Official Review · Reviewer_BHdp · 2023-11-06

**Soundness:** 2 fair
**Presentation:** 2 fair
**Contribution:** 2 fair
**Rating:** 3
**Confidence:** 5

**Summary:**

In this paper, the authors introduce DFedSGPSM, a DFL algorithm that leverages asymmetric topologies and the Push-Sum protocol for consensus optimization. They also use the SAM optimizer and local momentum to reduce local heterogeneity and speed up the optimization. They show that the algorithm converges in non-convex smooth settings and that the convergence rate depends on the connectivity of the topology. They propose a variant of the algorithm, DFedSGPSM-S, that gives clients more flexibility in choosing their neighbors for communication. They test the algorithm on standard datasets and show its advantage over a few other methods.

**Strengths:**

The paper tackles a crucial problem of decentralized optimization with asynchronous communications over directed networks. This problem has many applications and challenges for more realistic optimization frameworks. The authors use the Push-Sum protocol to design an algorithm and analyze its convergence behavior and implications. They also evaluate the algorithm on standard vision task and show its numerical performance.

**Weaknesses:**

The paper has some issues regarding the novelty and significance of the results. Firstly, the problem of asynchronous decentralized optimization has been already studied in [1,2,3] under a more general communication setting that accounts for message loss and delays. They also use the Push-Sum algorithm as the baseline and propose an algorithm with milder assumptions (removing bounded gradient assumption) and better convergence properties. The paper does not compare or discuss its method with these works. Secondly, the paper uses the SAM cost function [4,5] as a surrogate for the original objective function $f_i(\cdot)$ but it is unclear how the convergence result of the paper relates to the original problem. Lastly, the paper needs to improve its presentation and clarity, especially regarding the SAM algorithm, which is not well explained or motivated. The paper should state the problem that it aims to solve (SAM) clearly and make the writing more readable.


[1] Spiridonoff, Artin, Alex Olshevsky, and Ioannis Ch Paschalidis. "Robust asynchronous stochastic gradient-push: Asymptotically optimal and network-independent performance for strongly convex functions." Journal of machine learning research 21.58 (2020).
[2] Olshevsky, Alex, Ioannis Ch Paschalidis, and Artin Spiridonoff. "Fully asynchronous push-sum with growing intercommunication intervals." 2018 Annual American Control Conference (ACC). IEEE, 2018.
[3] Toghani, Mohammad Taha, Soomin Lee, and César A. Uribe. "Pars-push: Personalized, asynchronous and robust decentralized optimization." IEEE Control Systems Letters 7 (2022): 361-366.
[4] Foret, Pierre, et al. "Sharpness-aware minimization for efficiently improving generalization." arXiv preprint arXiv:2010.01412 (2020).
[5] Behdin, Kayhan, and Rahul Mazumder. "Sharpness-aware minimization: An implicit regularization perspective." arXiv preprint arXiv:2302.11836 (2023).

**Questions:**

- The authors accommodate SAM [4,5] optimization technique in their method (Line 8 of Algorithm 1) to address the surrogate cost function. How does the convergence analysis of Theorem 1 relate to the original objective function $f(\cdot)$, given that the algorithm uses SAM to optimize a modified function $f^\mathrm{SAM}(\cdot)$?
- Please explain what DFedSGPM (row 2) in Table 2 is and how it differs from DFedSGPSM. It seems that this method is not mentioned in the paper.
- How do you ensure a fair comparison between FedAvg and your algorithm, since FedAvg is a server-based and synchronous method, while your algorithm is a peer-to-peer and asynchronous method? Please also clarify the settings for the other methods in Table 1.

[4] Foret, Pierre, et al. "Sharpness-aware minimization for efficiently improving generalization." arXiv preprint arXiv:2010.01412 (2020).
[5] Behdin, Kayhan, and Rahul Mazumder. "Sharpness-aware minimization: An implicit regularization perspective." arXiv preprint arXiv:2302.11836 (2023).

---

### Official Review · Reviewer_9LTG · 2023-11-11

**Soundness:** 3 good
**Presentation:** 3 good
**Contribution:** 3 good
**Rating:** 5
**Confidence:** 4

**Summary:**

This paper studied the problem of decentralized learning over networks with asymmetric links (i.e., directed graphs). To solve this problem, the authors proposed a method that integrates several techniques, including sharpness-aware minimization (SAM) and momentum. The authors showed that their algorithm achieves an $\mathcal{O}(1/\sqrt{T})$ convergence rate in a non-convex smooth setting. The authors also conducted experiments to verify their theoretical results.

**Strengths:**

1. This paper provides a solid theoretical analysis for their proposed algorithm for asymmetric decentralized learning.

2. The authors conducted extensive experiments to verify their theoretical results and show the efficacy of the proposed DFedSGPSM.

**Weaknesses:**

1. The novelty of this paper is limited.

2. The theoretical results are inadequate.

Please see the detailed comments and questions below.

**Questions:**

1. Decentralized learning over networks has been a well-explored problem, although asymmetric decentralized federated learning is relatively less studied. Even so, none of the techniques in this paper is new. The debiased technique for asymmetric decentralized learning, SAM optimizer, and momentum have all been used in the literature. In essence, this paper carefully combines three existing techniques. However, such a combination does not bring sufficient differences in proof and analysis compared to those in the literature.

2. Also related to the previous bullet, one new element in this paper is to extend [Chen et al. 2023] to non-convex settings. But it's well-known that such an extension from strongly convex to non-convex and federated settings doesn't have any major hurdle and can still largely follow the same proof framework. Also, the extension of the K-local step decentralized FL is not new, it has been studied by some references cited in this paper and many others. The bounded gradient assumption (Assumption 5) is also strong. Such an assumption is not needed in many convergence performance analyses. Can it be relaxed? Due to these reasons, the $\mathcal{O}(1/\sqrt{T})$ convergence rate result is well expected.

3. The biggest rationale for incorporating the SAM optimizer is to improve the generalization performance. However, this paper only proved the convergence performance of the proposed DFedSGPSM method. There is no theoretical analysis for the algorithms's generalization performance. Hence, there is no theoretical understanding regarding how much generalization performance gain can be obtained due to the use of SAM.